# Clinical Application of Ultra-High-Frequency Ultrasound

**DOI:** 10.3390/jpm12101733

**Published:** 2022-10-19

**Authors:** Anna Russo, Alfonso Reginelli, Giorgia Viola Lacasella, Enrico Grassi, Michele Ahmed Antonio Karaboue, Tiziana Quarto, Gian Maria Busetto, Alberto Aliprandi, Roberta Grassi, Daniela Berritto

**Affiliations:** 1Department of Precision Medicine, University of Campania “Luigi Vanvitelli”, 80138 Naples, Italy; 2Department of Anatomical, Histological, Forensic and Orthopedic Sciences, Sapienza University of Rome, 00185 Rome, Italy; 3Department of Orthopedics, University of Florence, 50121 Florence, Italy; 4Department of Clinical and Experimental Medicine, University of Foggia, 71122 Foggia, Italy; 5Department of Law, University of Foggia, 71100 Foggia, Italy; 6Department of Urology and Renal Transplantation, University of Foggia Policlinico Riuniti of Foggia, 71122 Foggia, Italy; 7Department of Radiology, Istituti Clinici Zucchi, 20900 Monza, Italy; 8Department of Precision Oncology, University of Campania “Luigi Vanvitelli”, 80138 Naples, Italy; 9Italian Society of Medical and Interventional Radiology (SIRM), SIRM Foundation, 20122 Milano, Italy

**Keywords:** ultrasound, ultra-high frequency, musculoskeletal, imaging

## Abstract

Musculoskeletal ultrasound involves the study of many superficial targets, especially in the hands, wrists, and feet. Many of these areas are within the first 3 cm of the skin surface and are ideal targets for ultra-high-frequency ultrasound. The high spatial resolution and the superb image quality achievable allow foreseeing a wider use of this novel technique, which has the potential to bring innovation to diagnostic imaging.

## 1. Introduction

Ultra-high frequency ultrasound (UHFUS) is a method with the advantage of examining very small structures and providing high-resolution images with a dynamic, real-time, and comparative, where possible, evaluation [1]. UHFUS was established in clinical imaging in the 2000s, with its predominant application in dermatology and angiology [2,3,4]. In 1979, Alexander and Miller first introduced US as a noninvasive technique to measure normal skin thickness, and, in the 1980s and 1990s, high-resolution ultrasonography (HRUS) was used for the noninvasive assessment of skin nodules and cutaneous diseases [5,6]. The notion of UHFUS is often misunderstood due to a lack of consensus regarding the cutoff of frequencies for the classification of “very high frequencies” and “ultra-high frequencies”. The conventional ultrasonography (CUS) technique refers to the use of probes with frequencies ranging from 10 to 15 MHz. Bhatta et al. [7,8,9,10] considered high-frequency US (HFUS) to involve frequencies >10 MHz, while Polanska et al. [7,8,9,10] selected 20 MHz as the cutoff for HFUS. Finally, Shung et al. [7,8,9,10] defined HFUS for probes with frequencies >30 MHz. UHFUS, with frequencies as high as 50–70 MHz and capable of resolutions as fine as 30 microns, could permit new diagnostic applications in superficial small parts’ examination. A variety of superficial targets within the first 1 cm of the skin surface could be imaged, including for dermatological evaluation, such as the assessment of skin layers, hair follicles, nail units, and vascular, musculoskeletal, intraoral, and small-parts applications [11,12,13,14]. Hayaski et al. carried out a description of their experience in the identification and localization of lymphatic vessels in patients with secondary lymphedema. UHFUS provided images with extremely high resolution, demonstrating new characteristics of the lymphatic vessels [15,16]. The introduction of UHFUS in the study of oral lesions is one of the biggest innovations in US imaging of the head and neck region. The available frequencies of 48 and 70 MHz can effectively image the superficial layers of the mucosa as well as lesions of the oral cavity [17,18,19,20,21]. In comparison to CUS, UHFUS has a superior spatial resolution, even with the limitation of a low penetration depth, which is within 1–3 cm. In fact, 48 MHz probes have a penetration depth of 23.5 mm, while 70 MHz probes allow the imaging of the first 10.0 mm below the scanning surface.

The study of vascularization can be accomplished with the combination of color Doppler, which allows the observation of the distribution and size of vessels, and the power Doppler, which is usually more sensitive for detecting slow flow.

More than others, the method depends on the operator, as the probes are extremely sensitive to the examiner’s hand movements. The literature on UHFUS is still evolving, but this technology seems to be the answer to several diagnostic limitations related to the need for high-resolution investigation for both normal anatomy and disease processes. In this paper, we present the role and potential clinical applications of UHFUS in musculoskeletal imaging based on our experience with a commercially available ultrasound system (Vevo MD; FUJIFILM VisualSonics, Amsterdam, The Netherlands): an ultra-high-frequency clinical ultrasound system equipped with a 48 to 70 MHz linear-array transducer. The system yields an image resolution up to 30 μm, which is equivalent to half a grain of sand, within the first 3 cm from the body surface.

Musculoskeletal evaluation with UHFUS includes, in both adult and pediatric patients, physiological and pathological disorders of superficial joints, tendons, and nerves, as well as presurgical planning and post-treatment follow-up and a guide for non-surgical interventional procedures [22,23].

## 2. Tendons

UHFUS encompasses wide applications, especially in the evaluation of hand anatomy and pathological disorders of the upper arm, including the study of pulleys, fascia, retinacula, and other superficial structures of the soft tissues (Figure 1, Figure 2 and Figure 3). Particularly concerning the flexor tendon pulleys, the added value of UHFUS becomes crucial due to the subtle thickness of these structures, with a potential role in the identification of traumatic injuries [24,25,26]. UHFUS can identify pulley ruptures, especially with the dynamic assessment of the tendon-to-bone distance in flexion against resistance. In the evaluation prior to hand surgery, the analysis of hand anatomy can become essential for preoperative planning, postoperative management, and follow-up [27]. Tenosynovitis is defined as hypoechoic or anechoic thickened tissue with or without fluid within the tendon sheath with possible signs of Doppler signals, which are seen in two perpendicular planes [14]. Both tendon disease and tenosynovitis are important features of rheumatoid arthritis (RA), and US represents an ideal tool for their investigation. Tenosynovitis is seen as a combination of synovial thickening within the tendon sheath and tendon sheath effusion [28,29,30,31,32,33,34,35,36].

## 3. Small Joints

US is used routinely in rheumatologic and traumatic disorders to evaluate the joints of the wrist, hand, ankle, and foot. The use of UHFUS in the small joints of the hand and foot can demonstrate alterations such as synovial hyperplasia, the presence of calcifications, osteophytes, and an increased vascular signal and can be used to assess the thickness, homogeneity, and sharpness of the articular cartilage of peripheral joints, which is of great help in clinical practice [28,29,30,31]. In this field, above all, pediatric patients could benefit from such small transducers with great resolution, and this technology is designed for the smallest of patients with the greatest details and resolution possible (Figure 4 and Figure 5).

Over the last 10 years, US research in rheumatology has been standardized for early diagnosis and therapy monitoring [32,33,34,35].

The term synovitis is used to indicate the presence of synovial hypertrophy with a power Doppler signal and joint effusion, either proliferative or exudative. Its quantification via grayscale US usually uses a semiquantitative scale with three levels of intensity, indicating mild, moderate, or marked synovial changes [28,29,30,31,32,33,34,35,36]. According to the OMERACT indications, synovial fluid is defined as abnormal hypoechoic or anechoic intra-articular material that is displaceable and compressible and that does not exhibit a Doppler signal.

Erosions appear on US as focal discontinuities in the bone cortex. US assessment can provide detailed imaging of the hyaline cartilage, identifying small cartilage abnormalities in patients affected by RA, especially when dynamic US is performed in flexion of the finger joints and extension. Effusion is defined in US as abnormal hypoechoic or anechoic intra-articular material that is displaceable and compressible, but it does not exhibit a Doppler signal [37]. In the evaluation of crystalline arthropathies such as gouty arthropathy, US allows the evaluation of intra- and para-articular tophaceous deposits as well as the typical double-contour collateral ligaments. In calcium pyrophosphate deposition disease, US allows the evaluation of crystal deposits as well as hyperechoic dots or an irregular line within the cartilage layer [37,38,39,40]. UHFUS with 50 MHz probes is able to accurately identify cartilage thinning, cartilage echogenicity, and subchondral bone.

### Collateral Digital Ligaments

UHFUS offers a detailed and magnified representation of collateral ligaments of the hand (Figure 6).

This could become useful in demonstrating even partial injuries of these structures. In particular, injury to the ulnar collateral ligament (UCL) complex of the thumb is a common traumatic lesion that requires prompt imaging evaluation for adequate treatment. UHFUS could have a key role in the illustration of both static and dynamic findings related to UCL injuries with even more details than MRI imaging (Figure 7 and Figure 8).

## 4. Nerves

Compared to 5–20 MHz probes, UHFUS equipped with 80 MHz probes allows a more detailed evaluation of nerve anatomy in small cutaneous nerve branches, giving a spatial resolution down to 30 microns [6,8,10,41,42,43] (Figure 9).

UHFUS allows the precise evaluation of the normal anatomical appearance as well as the echo-structural alteration of the honeycomb appearance represented by hypoechoic fascicles on a hyperechoic background. Entrapment neuropathies are generally characterized by nerve thickening, the loss of the fascicular pattern, and decreased echogenicity, mainly due to edema (in the acute phase) and fibrosis (in the chronic phase). It is also possible to see the thinning of the nerve at the site of compression. Conversely, in other disorders, such as Charcot–Marie–Tooth and some nerve neoplasms, we can see the enlargement of selected fascicles, an imaging feature recognizable using UHFUS [44,45,46]. UHFUS allows the determination of the location, size, and type of lesion, nerve swelling, and inflammation or the identification of compressive structures such as calcifications and scar tissue [47,48]. Nerves that are often involved in traumatic and iatrogenic injuries are the superficial cutaneous branches of the median, ulnar, and radial nerves in the hand and wrist and the digital nerves. UHFUS allows the visualization of the small cutaneous branches of peripheral nerves. The advantages of using US in the evaluation of the anatomy and disorders of the median nerve even after surgery are known in the literature: in cases of post-traumatic neoformations, such as a neuroma, both intra- and extra-operative applications are used for the preservation of the nerve after surgical excision. UHFUS could be useful in the study of pediatric patients and in neuromuscular disorders such as chronic inflammatory demyelinating diseases, thanks to the evaluation of the number of morphologies and the size of the nerve fascicles [49,50,51]. Benign lesions of the peripheral nerves, such as a lipofibromatous hamartoma and the infiltration of the nerve fascicles, appear as non-homogeneous nerve enlargements, especially in proximal median nerve segments and the brachial plexus [43,52], with variable echogenicity patterns depending on the stage of the disease. UFUS has been used for non-surgical guidance in percutaneous procedures for partial and total wrist denervation. In the treatment of chronic wrist pain due to post-traumatic injuries, degenerative disorders, or arthritis, denervation is the treatment for chronic wrist pain, without impairing motor function, and avoids the need for postoperative immobilization to decrease the risk of stiffness. Under US guidance, the dorsal extensor tendon compartment is visualized, and, using a transverse anatomic projection along the short axis of the tendons, radiofrequency ablation of the posterior and anterior interosseous nerves is performed. The posterior interosseous nerve appears as a thin, hypoechoic, and non-compressible ovoid structure from 1 to 3 mm in diameter (Figure 10) [53,54,55].

## 5. Soft-Tissue Masses

UHFUS could represent an advantageous technique for the identification of small and soft-tissue neoformations such as glomus tumors (Figure 11 and Figure 12).

Glomus tumors are rare, benign, vascular neoplasms arising from the glomus body, which is a contractile neuromyoarterial structure found in the reticular dermis. This structure controls blood pressure and temperature by regulating blood flow in the cutaneous vasculature. Hyperplasia in any of these parts can lead to tumor formation, which is extremely painful. Glomus tumors account for 1–5% of soft-tissue tumors of the hand, and 75% of them are subungual in location. Other less commonly involved sites in the hand are the nail matrix, nail bed, and pulp of the finger. The delay in diagnosing these tumors for many years is a significant problem. It is not uncommon that patients are easily misdiagnosed with conditions such as neuropathic complaints, arthritis, or neuralgia and undergo unsuitable treatment. For these reasons, when the clinical examination is equivocal, noninvasive imaging techniques may be needed to aid in the diagnosis and delineate the anatomy preoperatively. Complete surgical excision of the tumor is the only effective treatment. Incomplete excision is considered the main cause of recurrence. US follow-up and/or intraoperative US may be useful for reducing recurrence and ensuring adequate resection, and the UHFUS tool is promising in this field [56].

## 6. Conclusions

Musculoskeletal evaluation with UHFUS includes, in both adult and pediatric patients, physiological and pathological disorders of superficial joints, tendons, and nerve structures, as well as presurgical planning and post-treatment follow-up and a guide for non-surgical interventional procedures. The high spatial resolution and the superb image quality achievable allow foreseeing the wider use of this novel technique, which has the potential to bring innovation to diagnostic imaging.

## Figures and Tables

**Figure 1 jpm-12-01733-f001:**
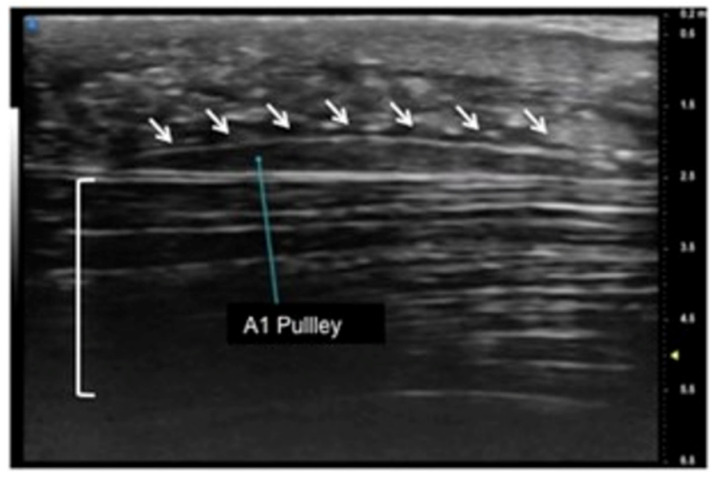
Sagittal view of A1 pulley (white arrows) presenting as a fusiform structure with a hypo-echoic signal contoured by a thin hyperechoic line. The superficial flexor tendon is visible (square parenthesis).

**Figure 2 jpm-12-01733-f002:**
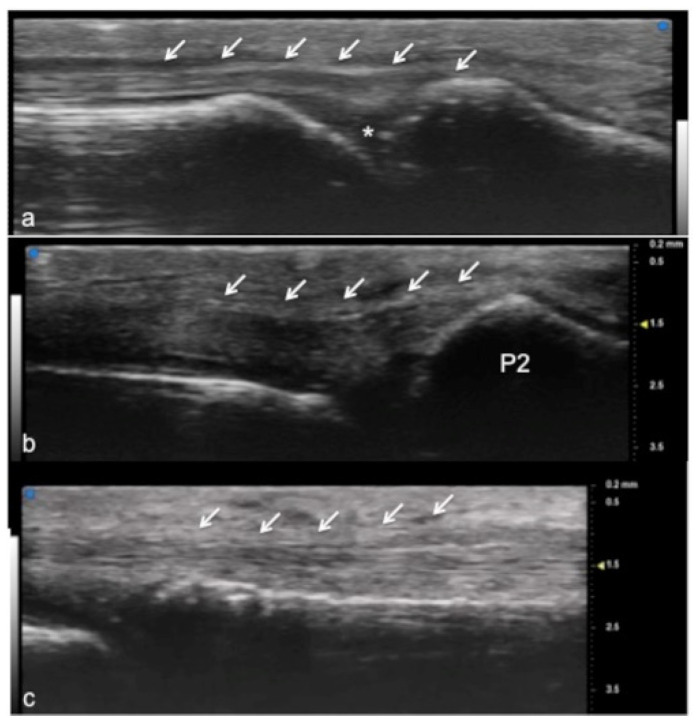
Extensor tendon of a finger. UHFUS gives a detailed and magnified representation of small structures such as the extensor tendon of the finger, even allowing the visualization of partial lesions. In (**a**), the sagittal view of the terminal extensor tendon (white arrows) at the level of the distal interphalangeal joint (white asterisk). In (**b**), the median band of the extensor tendon inserting at the level of the middle phalanx (P2). In (**c**), the thin sagittal band of the extensor tendon at the level of the metacarpal head.

**Figure 3 jpm-12-01733-f003:**
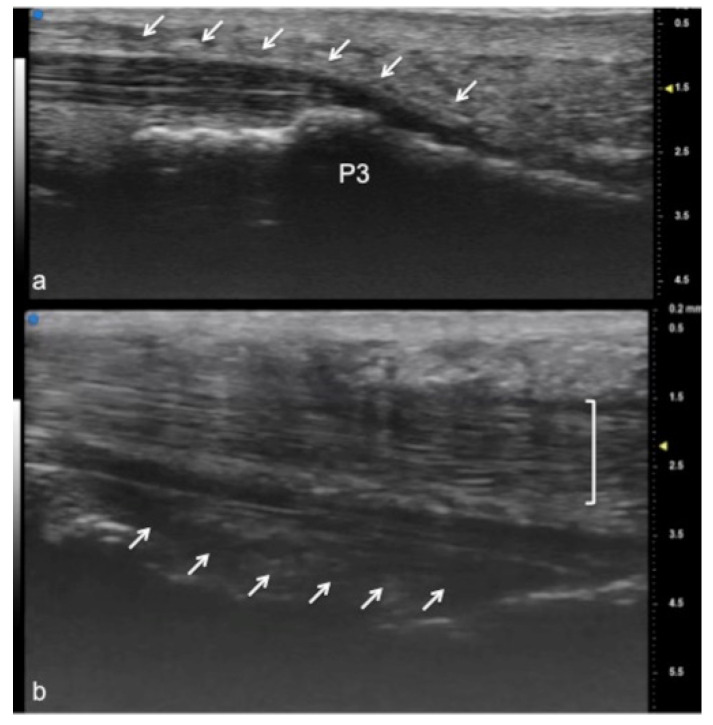
Flexor tendon of a finger. Using UHFUS at the flexor tendons, the same spatial resolution can be achieved as in the imaging of the extensor tendons. In (**a**), the sagittal view of the deep flexor tendon component inserting on the basis of the distal phalanx (P3). In (**b**), the superficial component of the flexor tendon (white arrows) lying near the deep component (white square parenthesis).

**Figure 4 jpm-12-01733-f004:**
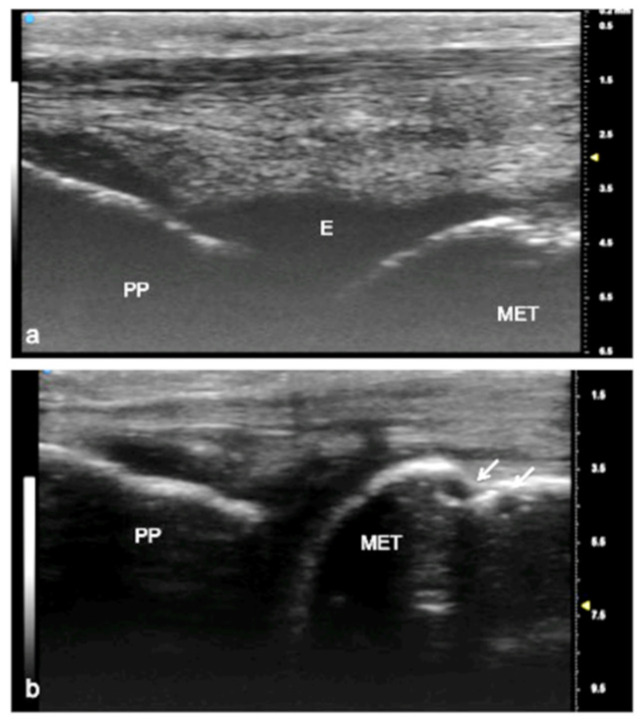
Juvenile idiopathic arthritis (JIA). UHFUS gives clear details of pathological findings in pediatric patients with JIA. In (**a**), articular effusion (E) at the level of the metacarpal–phalangeal joint (MET-PP). In (**b**), osseous erosions at the level of the metacarpal head (white arrows).

**Figure 5 jpm-12-01733-f005:**
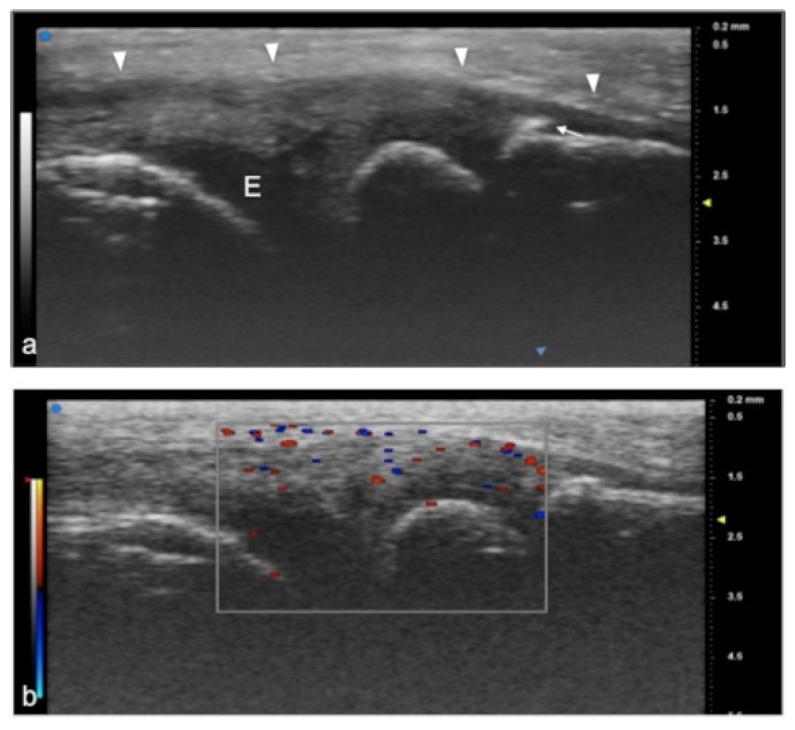
Juvenile idiopathic arthritis. UHFUS gives clear details of pathological findings in pediatric patients with JIA. In (**a**), articular effusion (E) at the level of wrist and enthesophyte (white arrow). Thickened capsule (arrowheads) in (**a**) presenting with an increased Doppler signal (white square) in (**b**).

**Figure 6 jpm-12-01733-f006:**
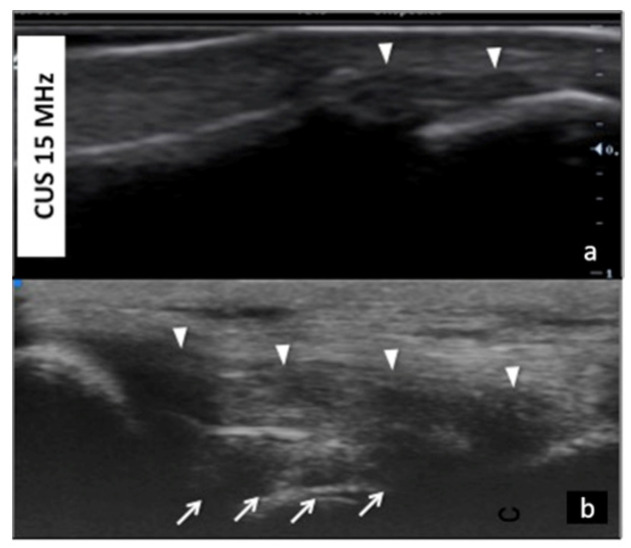
Collateral ligaments of interphalangeal joints: comparison between 50 MHz probes in (**a**) and 15 MHz CUS (arrowheads) in (**b**). In (**a**), UHFUS gives a more detailed and magnified representation of both deep (white arrows) and superficial (white arrowheads) components of the ligament.

**Figure 7 jpm-12-01733-f007:**
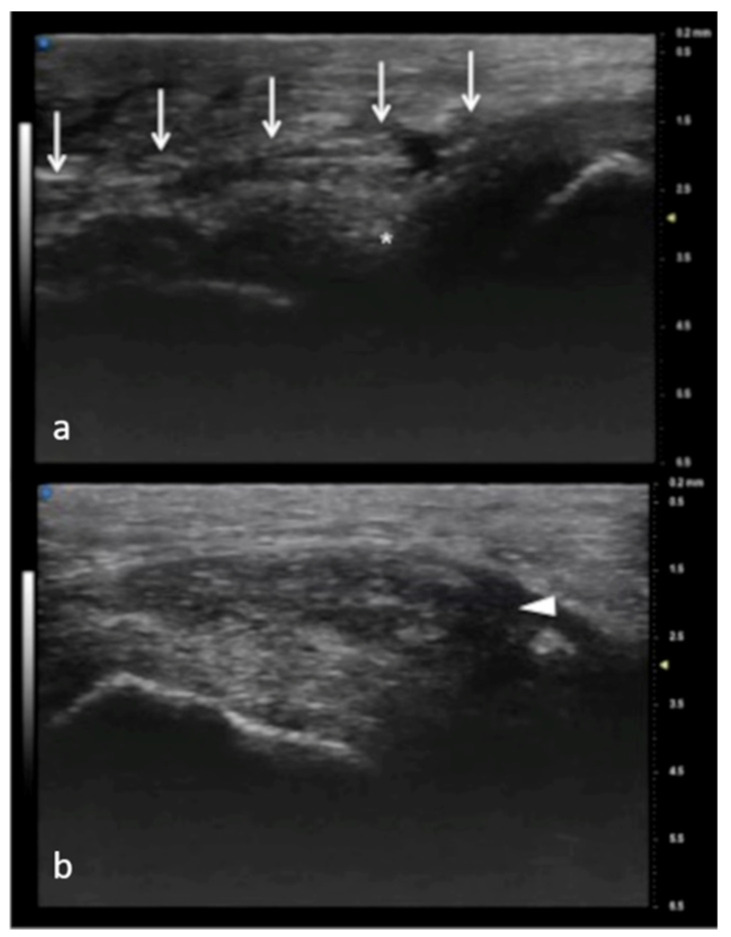
Full-thickness UCL tear. In (**a**), normal appearance of adductor pollicis aponeurosis (white arrows) and below the ulnar collateral ligament (asterisk) of the first metacarpophalangeal joint. In (**b**), the non-visualization of the ulnar collateral ligament and the presence of a mass-like area (arrowhead) proximal to the joint have high accuracy in depicting a displaced full-thickness ligament tear.

**Figure 8 jpm-12-01733-f008:**
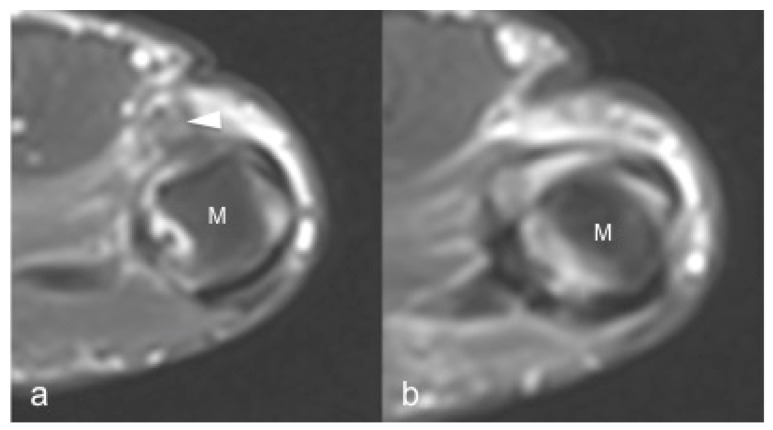
Same patient as in Figure 7. 1.5 T MRI shows a mass-like area (arrowhead) proximal to the joint representing the injured and displaced ligament in (**a**). In (**b**), the contralateral metacarpal phalangeal joint of the thumb. M: metacarpal head.

**Figure 9 jpm-12-01733-f009:**
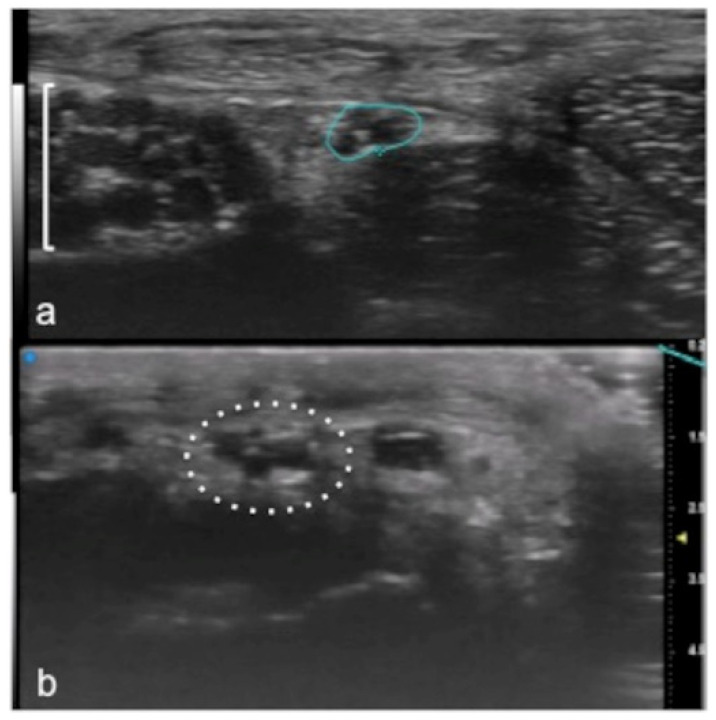
Visualization of small peripheral nerves using UHFUS. Recurrent branch (line) of the median nerve (square parenthesis) at the level of the thenar eminence in (**a**). Digital branch (dot circle) of the median nerve at the level of the finger in (**b**).

**Figure 10 jpm-12-01733-f010:**
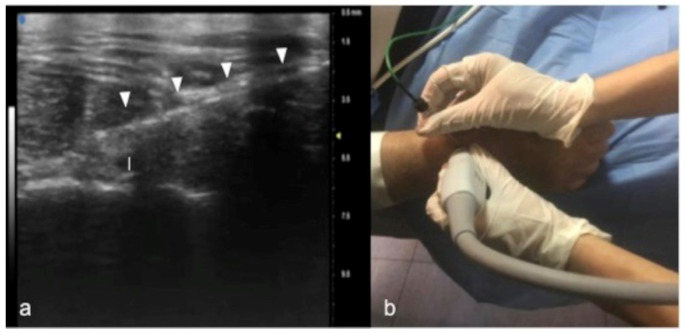
Percutaneous radiofrequency ablation of the posterior interosseous nerve for chronic wrist pain: In (**a**), cannula’s insertion (white arrowheads) under real-time ultrasound. Guidance for direct visualization of the posterior interosseous nerve (white caliber). The procedure is performed on an awake patient using a noninvasive approach (**b**).

**Figure 11 jpm-12-01733-f011:**
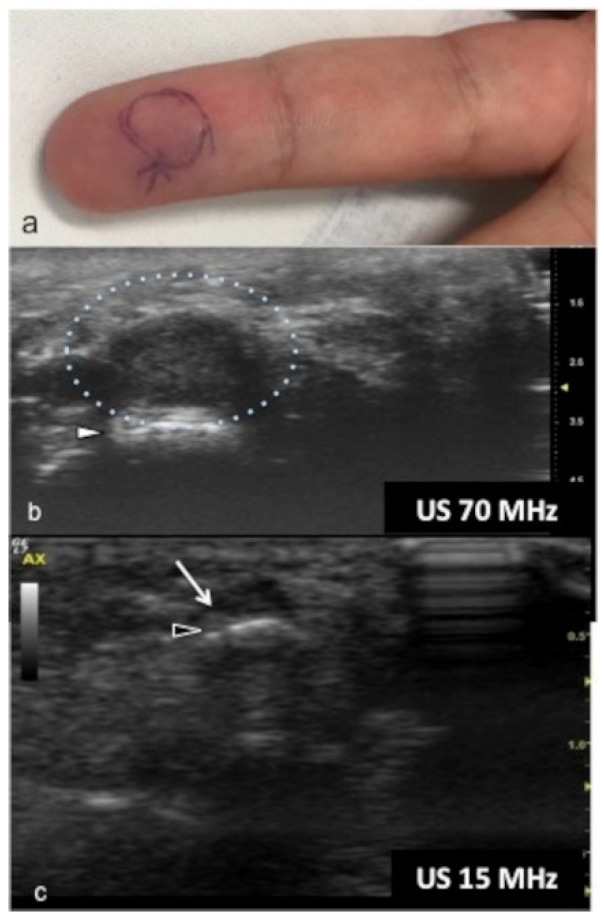
Glomus tumor. The location corresponds to the pulp of finger IV (**a**). The patient had the classical triad of symptoms: paroxysmal pain, pinpoint pain, and cold hypersensitivity, lasting for two years. In (**b**), UHFUS shows a well-delimited nodule in contact with the adjacent phalangeal bone (arrowhead), but no cortical deformity is present. No significant hyperemia on color Doppler was noticed. On CUS (**c**), the nodule was delineated only thanks to the help of the preliminary UHFUS exam.

**Figure 12 jpm-12-01733-f012:**
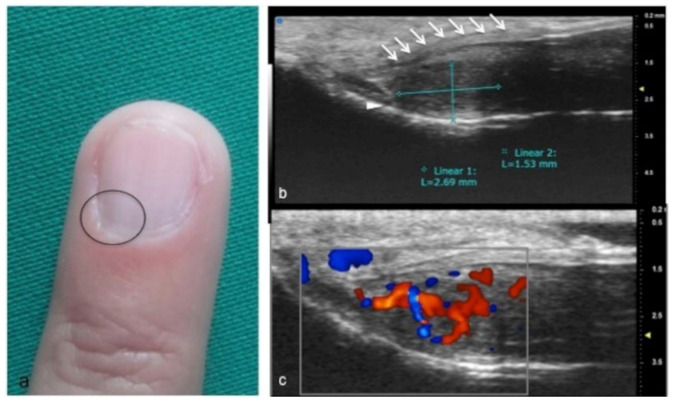
Glomus tumor in the subungual space. No clear alterations were visible during clinical examination at the site of pain (circle in (**a**)). In (**b**), UHFUS effectively demonstrates the presence of a hypo-isoechogenic nodule (calibers) in contact with the adjacent phalangeal bone (arrowhead) under the nail plate (arrows). Note the small deformation of the nail plate. Mild vascularization on color Doppler was present (**c**).

## Data Availability

Not applicable.

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
