# Peer review of "Clinical Application of Ultra-High-Frequency Ultrasound"

_jpm, 2022, doi:10.3390/jpm12101733_

Round 1

Reviewer 1 Report

Dear Authors.

Your paper gives a good overview on the UHFUS. Of course, this is not a scientific evaluation, but a review. 

I would like to ask you for a short addition, namely on the use of elastography in UHFUS.

I have made some stylistic suggestions to improve the English text (see PDF file).

Best regards

Author Response

Dear Reviewer, 

thank you for your suggestions.

The manuscript has been updated following your corrections.

Regarding the use of elastography, authors gave no mention about since ultra-high frequency ultrasound (50-80 mhz) is not equipped with this tool.

Thank you for your availability.

Regards  

Reviewer 2 Report

Thank you for your excellent article. I read it with pleasure. Here are some of my comments.

1. [113-116] The E structure marked in Figure 5 (a) seems to be the normal cartilage tissue of the wristbone and is not an effusion. Also marked "thickened capsule" by arrowheads are the hypoechogenic extensor tendons due to anisotropy. Finally, in part B, the Doppler window presents diffuse color artifacts rather than the increased blood flow in the capsule. In summary - there is the normal dorsal child's wrist presented (except for suspicious erosion marked with an arrow)

2. [140] I would very much ask to turn a pre-clockwise look, maintaining the same ultrasound image style in the article. In addition, the rotated ultrasound image doesn't provide any additional information, mainly when the ultrasound images are not evaluated in such a presentation during everyday practice.

3. [150-153] ]In Figure 7, both parts are not separated into a and b. In addition, the presentation of the Stener is not clearly visible in the second part. There are no marked appropriate bones. In summary - Figure 7 is not representative of the Stener lesion. It would be good to change to better quality ultrasound images.

4. [159-160] I'm afraid I have to disagree with the author's statement about CUS and its inability to evaluate tiny nerves with 5-20 MHz probes. From personal practice with a 12 MHz ultrasound probe, it is possible to see the small finger nerves, even to the middle phalanxes or the palmar cutaneous and recurrent branches of the median nerve. I would suggest presenting this argument less categorically.

5. It would be reasonable if the authors still mention the capabilities of blood flow evaluation using UHFUS

Author Response

Dear Reviewer, following your instructions, an updated version of the manuscript has been attached.

Thank you for your availability

Regards
